# Relationship between Locomotive Syndrome and Musculoskeletal Pain and Generalized Joint Laxity in Young Chinese Adults

**DOI:** 10.3390/healthcare11040532

**Published:** 2023-02-10

**Authors:** Yixuan Ma, Xinze Wu, Shaoshuai Shen, Weihao Hong, Ying Qin, Mingyue Sun, Yisheng Luan, Xiao Zhou, Bing Zhang

**Affiliations:** 1Division of Sports Science and Physical Education, Tsinghua University, Beijing 100081, China; 2Department of Internal Medicine and Rehabilitation Science, Tohoku University Graduate School of Medicine, Tohoku University, Sendai, Miyagi 980-8574, Japan; 3School of Education and Welfare, Aichi Prefectural University, 1522-3 Ibaragabasama, Nagakute, Aichi 480-1198, Japan; 4Department of Physiotherapy, Planet Rehabilitation Center, Planet Rehabilitation Technology Co., Ltd., Guangzhou 510623, China; 5School of Physical Education, Huazhong University of Science and Technology, Wuhan 430074, China

**Keywords:** locomotive syndrome, musculoskeletal symptoms, young adults

## Abstract

This study aims to investigate the prevalence of locomotive syndrome (LS) and to examine the relationship of LS with musculoskeletal symptoms (pain, generalized joint laxity (GJL)) in young Chinese adults. Our study population (n = 157; mean age of 19.8 ± 1.2 years) comprises college student residents at Tsinghua University in Beijing, China. Three screening methods were used to evaluate LS: 25-question Geriatric Locomotive Function Scale (GLFS-25), a two-step test, and a stand-up test. Musculoskeletal pain was assessed by self-report and visual analog scale (VAS), and joint body laxity was evaluated using the GJL test. The prevalence of LS was 21.7% of all participants. Musculoskeletal pain affected 77.8% of the college students with LS and was strongly associated with LS. A total of 55.0% of college students with LS had four or more site joints that were positive for GJL, and higher scores of GJL were associated with a higher prevalence rate of LS. Young Chinese college students have a relatively high prevalence of LS, and musculoskeletal pain and GJL were significantly related to LS. The present results suggest that we need early screening of musculoskeletal symptoms and LS health education in young adults to prevent the mobility limitations of LS in the future.

## 1. Introduction

China has rapidly advanced into a super-aging society in recent years. By 2020, 13.5% of the Chinese population were older than 65, approximately 190 million people [1]. Since various chronic diseases are gradually trending in youths [2], health interventions from youth are urgently needed, given the rising geriatric problem. The Japanese Orthopaedic Association (JOA) proposed the term “locomotive syndrome” (LS) in 2007, which is defined as a decline in mobility or quality of life due to the dysfunction of the motor system, including muscles, bones, joints, spinal cord, and peripheral nerves [3,4]. However, LS affects not only middle and older adults, whose incidence of LS is approximately 21.1% and 49.3% [5], respectively, but also several people in their 20s meet the LS definition [6,7,8,9]. However, studies on the prevalence of LS in young people are still scarce. Since the decrease in mobility and muscle strength leading to disability may start at a young age [4,8,10,11], it is necessary to pay more attention to the early screening and prevention of LS in young people [9,12,13].

Some studies have found that the risk of LS is associated with muscle mass, lower limb muscle exertion [9,12], BMI [6], physical activity [7,14], and walking speed [5,12] in young adults. Previous studies claimed that older patients with LS frequently experience musculoskeletal pain in multiple sites [15,16,17]. However, few studies have focused on young people [7,18], but the percentage of young adults with patellofemoral and knee pain was reported to be 20.7% and 35.6%, respectively [19]. In addition, a connective tissue disorder known as generalized joint laxity (GJL) [20] causes musculoskeletal symptoms and physical limitations, particularly in adolescents [21,22]. Therefore, starting LS preventive measures in young adults requires attention to body pain and joint laxity.

There are two main purposes of this study: (1) to determine the prevalence of LS and (2) to investigate the relationship between LS and musculoskeletal symptoms (e.g., pain, GJL) in young Chinese college students.

## 2. Materials and Methods

### 2.1. Participants

The participants of this study were recruited from June to August 2022, and data were collected from September to November 2022 at Tsinghua University, Beijing, China. The recruitment methods included oral communications in physical education (PE) classes and fielding responses from posters on campus for three months. Participants were excluded if they had walking function problems and could not participate in all assessments of LS and GJL. All participants were free from chronic disease (e.g., diabetes, cardiovascular) and musculoskeletal injuries (e.g., fracture, orthopedic surgery) as assessed by their annual check-ups. There were no professional college athletes, and no one participated in long-term professional or intensive sports training prior to this study. The data analyses included 157 students (mean age, 19.8 ± 1.2 years) who met the criteria. This research was approved by the Ethics Committee at Tsinghua University, and the methods were carried out in accordance with the principles of the Declaration of Helsinki.

### 2.2. Measurement and Measuring Equipment

#### Physical Measurement

Physical measurements included height and body composition analysis, which adapted direct segmental multifrequency bioelectrical impedance analysis (BIA) (InBody720; Biospace Co, Ltd., Seoul, Korea). The BIA provided absolute values for body weight, skeletal muscle mass (SMM), appendicular skeletal muscle mass (ASM), body fat mass (PBF%), and total body water (TBW). Grip strength (kg) was measured using a handheld dynamometer (GRIP-D; Takei Ltd., Niigata, Japan). Both hands were tested, and the higher value was used to characterize the maximum muscle strength of the subject. Gait speed was assessed with the usual gait speed, and balance was measured using the 5-times sit-to-stand test (5xSST). To measure gait speed, two photocells were placed 6 m apart to record the time participants could walk at their usual speed, and the average speed of two walks was recorded. In the 5xSST, participants were instructed to sit against the back of a chair. Participants were asked to stand up and sit down as quickly as possible 5 times with their arms crossed in front of their chest. A chair with armrests and a standard height of 0.47 m was used.

### 2.3. Locomotive Syndrome (LS) Risk Tests

The “stand-up test,” “two-step test,” and “the 25-question geriatric locomotive function scale” (GLFS-25) were components of the LS risk test [4,5,12] recommended by the JOA, which also applies to young adults [13].


**Stand-up test**


The stand-up test measures lower muscle strength by having the participants stand on one or both legs from a 40, 30, 20, or 10 cm–high seat. The details of the testing method and scoring criteria were based on previous studies [8,12], and a participant‘ successful performance was given a score between 0 and 8.


**Two-step test**


(a) Participants stood with both feet behind the starting line; (b) participants were instructed to align their feet after taking two extremely long steps; (c) the length of the two steps between the starting line and the tips of the participant’s toes was measured. The following formula was used to determine the score for the two-step test: length of the two stages (cm) ÷ height (cm).


**GLFS-25**


It is a self-administered, comprehensive measure consisting of 25 items that include four questions regarding pain during the last month [23], 16 questions regarding activities of daily living during the last month, 3 questions regarding social functions, and 2 questions regarding mental health status during the last month. These 25 items were graded on a five-point scale, from no impairment (0 points) to severe impairment (4 points), and then arithmetically added to produce a total score.

We defined LS as follows based on the results of the three tests:

**No-LS:** If all three conditions were met—(a) two-step score ≥1.3; (b) ability to stand up on a single leg from a 40 cm–high seat with each leg; and (c) GLFS-25 score <7. **LS 1:** If any of the three conditions (a–c) were met—(a) stand-up test, difficulty in standing from a 40 cm high seat using one leg (either leg); (b) two-step test <1.3; or (c) 25-question score ≥7). **LS 2:** If any of the three conditions (a–c) were met—(a) stand-up test, difficulty in standing from a 20 cm high seat using both legs; (b) two-step test <1.1; or (c) 25-question score ≥16).

### 2.4. Musculoskeletal Pain

Musculoskeletal pain was defined by asking a question about nine anatomical sites (neck, shoulder, upper limb, lumbar, hip, knee, ankle, foot, and other): Have you experienced pain on most days (that continue on at least one day) in the past month in addition to the current pain? Musculoskeletal pain was deemed present in an anatomical site for participants who answered “yes.” [7,24]. Furthermore, we used the pain visual analog scale (VAS) [18] to evaluate the pain degrees.

### 2.5. General Joint Laxity Test

Seven maneuvers of the University of Tokyo GJL test were performed [25]. The judgment criteria of the positive joint laxity were the following: (1) wrist: passive opposition of the thumb to the flexor aspect of the forearm; (2) elbow: hyperextension >15 degrees; (3) shoulder: fingers overlap or grasping; (4) knee: hyperextension >10 degrees; (5) ankle: dorsiflexion >45 degrees; (6) spine: trunk flexion with the knee extended and both palms contacting the mat; and (7) hip: toes pointing outwards >180 degrees [26] (Figure 1). Seven positions were measured, with one point being given to each item; except for the spine, the left and right positions were each given a value of 0.5 points for the six major bilateral joints. A goniometer was used to measure the items with the joint angle as the criterion. One operator was responsible for taking and recording joint angle measurements.

Demographic variables and behavioral characteristics included age, gender, smoking (current smoker, ex-smoker, non-smoker) and drinking habits (drinks every day, drinks occasionally, ex-drinker, never drinks), and the International Physical Activity Questionnaire (IPAQ). All parts of the tests were completed by professionally trained medical staff. Part of the questionnaire test was conducted via a face-to-face interview.

### 2.6. Statistical Analysis

The anthropometric characteristics and background data with continuous variables were expressed as mean ± SD and categorical variables as absolute numbers and percentages (%) of the total. Differences between variables were examined using ANOVA with Bonferroni correction (continuous variables) or using the chi-square test (categorical variables). Logistic regression analyses were used to assess the association between the LS group, the No-LS group and pain sites, pain VAS and GJL sites, and GJL score. Model 1 was adjusted for age and sex. Model 2 was adjusted for Model 1 covariates plus BMI, grip strength/weight, ASMI, PBF%, GJL score (or total pain VAS), smoking, drinking, and IPAQ. The odds ratio (OR) value and 95% CI were calculated. All statistical analyses were performed using SPSS v 26.0 (SPSS Inc., Chicago, IL, USA), and *p*-values of <0.05 were considered statistically significant.

## 3. Results

Anthropometric characteristics and background data of all subjects, the No-LS group, and the LS group are shown in Table 1. The mean age of the 157 participants was 19.8 ± 1.2 years, and 57 participants (36.3%) were female. The prevalence of LS was 21.7% of all participants (16.0% of males (LS1: 13.0%, LS2: 3.0%) and 31.6% of females (LS1: 28.1%, LS2: 3.5%)). The prevalence of a two-step test score <1.3, the inability to stand with one leg from a 40 cm high seat, and a GLFS-25 score ≥7 were 0.6%, 2.5%, and 18.5%, respectively. The total number of pain sites, pain VAS score, GJL score, PBF%, and GLFS-25 score were significantly higher in the LS group than in the No-LS group. The LS group had a significantly lower stand-up test score, ASMI, and adjusted grip strength.

Figure 2 shows the prevalence of musculoskeletal pain in the nine anatomical areas, and 57.3% of persons had experienced musculoskeletal pain somewhere in their bodies. The typical pain sites in the LS group were the neck (*p* = 0.021), shoulder (*p* = 0.002), lumbar (*p* = 0.007), knee (*p* = 0.004), foot (*p* = 0.002), and other sites (*p* = 0.009). After the multivariate analysis (Table 2), the prevalence of pain in shoulder (OR = 2.99, 95% CI = 1.18–7.59, *p* = 0.021), lumbar (OR = 2.93, 95% CI = 1.19–7.21, *p* = 0.019), knee (OR = 3.19, 95% CI = 1.16–8.73, *p* = 0.024), and foot (OR = 6.18, 95% CI = 1.95–19.6, *p* = 0.002) were significantly higher in the LS group than in the No-LS group. Furthermore, the VAS of the total pain (OR = 1.13, 95% CI = 1.05–1.22, *p* = 0.002), neck pain (OR = 1.57, 95% CI = 1.13–2.19, *p* = 0.008), shoulder pain (OR = 1.67, 95% CI = 1.15–2.42, *p* = 0.008), ankle pain (OR = 1.80, 95% CI = 1.04–3.11, *p* = 0.034), and foot pain (OR = 3.55, 95% CI = 1.47–8.56, *p* = 0.005) were significantly higher in the LS group than in the No-LS group after the multivariate analysis (Table 3). There were no differences in GJL sites between the LS group and the No-LS group, except for the elbow in unadjusted crude (OR = 2.48, 95% CI = 1.14–5.38, *p* = 0.022) (Table 4 and Figure 3). However, after the multivariate analysis, the GJL score was significantly higher in the LS group than in the No-LS group, and the risk of LS increased with the increase of the GJL score (OR = 1.42, 95% CI = 1.06–1.91, *p* = 0.019) (Table 5).

## 4. Discussion

The main findings of this study are as follows: First, we investigated the prevalence of LS in young college students (mean age of 19.8 ± 1.2 years). The overall prevalence of LS was 21.7% (16.0% of males (LS1: 13.0%, LS2: 3.0%) and 31.6% (LS1: 28.1%, LS2: 3.5%) of females). Second, musculoskeletal pain affected 77.8% of the college students with LS. Musculoskeletal pain was more common in the LS group than in the No-LS group and was strongly associated with LS. Third, 55.0% of college students with LS had four or more site joints that were positive for GJL (No-LS group: 29.3%), and higher scores of GJL were associated with a higher prevalence rate of LS.

### 4.1. Prevalence of LS in Young Adults

A few previous studies from Japan have investigated the incidence and risk propensity of LS in young people [6,8,9,12,13]. Akinobu et al. [13] researched the prevalence of LS among young workers under 29 years of age to be 19.6% in men and 5.3% in women. In addition, the prevalence was higher in men <29 years old than in those 30–39 and 40–49 years old. Another study [9] found that the prevalence of LS among young female students was 18.5%, while the prevalence in males was 0% (mean age of 18.6 years), which was lower than that determined by the present study. The main reason for the difference was that our participants had relatively high GLFS-25 scores, with approximately 18.5% of them ≥7 [5]. Another possible reason was that the participants (especially males) in the present study had higher BMIs (21.9 kg/m^2^ vs. 20.4 kg/m^2^) [6] and PBF% (17.3% vs. 13.2%) [12] and lower muscle strength (35.2 kg vs. 42.1 kg) [9], which are significant risk factors for LS in young adults. Moreover, because of the increase in online classes and outdoor activity limitations during the novel coronavirus disease 2019 (COVID-19), the participants in this study had more sedentary time compared to previous studies [27,28], which might also contribute to a relatively high prevalence in our study [7,29].

In addition, one study found that 65.0% of female college students (20.0 ± 1.5 years) were classified in the high LS risk group [12] based on the JOA-recommended cut-off values (one leg stand of 30 cm or less, two-step test <1.55) for those in their 20s [8]. In this study, 56.0% of female students were assigned to the LS risk group after using the same cut-off values. A study found that the stand-up and two-step tests might start to deteriorate in those in their 30s and younger, which might support the high rate of LS risk in young people [5]. However, the majority of the studies mentioned above were from Japan, and the differences in ethnicity and lifestyle [30] might also contribute to different prevalence.

### 4.2. Musculoskeletal Pain and LS

Several previous studies clarified that musculoskeletal disorders such as lower extremity pain [16,17], multiple sites pain [31], osteoporosis [32], and lumbar spinal stenosis [16] are associated with LS in older people. However, few studies focused on young adults, especially those under 30 [7,33]. In the present study, 77.8% of the college students with LS had experienced musculoskeletal pain. The incidence of pain in the shoulders (22.1% of No-LS and 50.0% of LS), lumbar (23.0% of No-LS and 47.1% of LS), knee (13.1% of No-LS and 35.3% of LS), and foot (8.2% of No-LS and 29.4% of LS) were significantly higher in the LS group than in the No-LS group (Table 2), consistent with previous research in young and middle-aged adults [7]. Furthermore, Hironori et al. [17] revealed that 85.8% of patients over 40 with chronic pain were diagnosed with LS2, indicating that chronic pain patients had an earlier risk of developing LS.

Moreover, the students with LS had more pain sites in the lower extremities than upper extremities, indicating the pain origin of LS was induced by the degeneration of the lumbar disk [34] and mainly affected mobility functions, such as gait and climbing ability [4,5]. In addition, because the spine and knee joint are primarily supporting the person’s upright body and trunk, lower extremity pain is the primary cause of locomotive disability [35]. Multivariate analysis revealed the pain degree (total pain of VAS, OR = 1.13, *p* = 0.002) was significantly worse in the LS group than in the No-LS group in our study. This result was similar to Takeshi et al.’s [18] finding that pain VAS was the risk factor for LS in patients with rheumatoid arthritis (OR = 1.04, *p* < 0.001). In addition, as an evaluation of locomotive organ deterioration, GLFS-25 was found to be associated with chronic pain [17,32]. In contrast, our results showed a relatively higher score of GLFS-25 (mean value: 4.0 ± 4.9) in young adults [5]; in particular, questions about pain review scored higher. According to the current findings, LS was associated with musculoskeletal pain even in college students, as in older adults with LS.

### 4.3. GJL and LS

GJL has a high prevalence among adolescents (especially in women) and declines with age [25,36]. Previous studies have shown that GJL is a risk factor for musculoskeletal pain, lower extremity joint injuries [37], increased plantar loading [20], and proprioception deficits [21,38]. In the present study, we found the GJL score was significantly higher in the LS group than in the No-LS group (OR = 1.42, *p* = 0.019); 55.0% of college students with LS had four or more site joints that were positive for GJL. Two possible mechanisms explain this result. One is that LS is mainly associated with lower extremity function, and the range of motion is greater than normal in GJL, which may affect dynamic balance and alignment of the lower extremities [20]. Second, knee hyperextension results in impaired proprioception [39] and lower muscle strength [22], which can induce knee injury and inadequate control of lower motor function [38]. For instance, 20.6% of college students with LS had knee hyperextension >10 degrees in our study. Hence, a lack of awareness about the relationship between GJL and LS may lead to unnecessary motor system dysfunction in young college adults.

### 4.4. Strength and Limitations

This study was the first cross-sectional study to identify the prevalence of LS in young Chinese adults; we clarified that musculoskeletal pain and GJL at a young age are associated with LS.

Several limitations must be considered in this study. Firstly, the current findings were only representative of some Chinese college students because they were from one university, and most were first-year students. Secondly, there was a small sample size; additional studies with large sample sizes and robust experimental designs should be carried out, including verifying the reliability and validity of the GLFS-25 in young adults. Thirdly, our data did not include the medical examinations for bone mineral density, scoliosis, etc., and a more standard GJL assessment method, such as the Beighton test, should be supplemented in future research.

## 5. Conclusions

In conclusion, this cross-sectional study aimed to identify the prevalence of LS and its association with musculoskeletal pain and GJL in young Chinese college students. The overall prevalence of LS was 21.7% (16.0% of males and 31.6% of females). Musculoskeletal pain was more common in the LS group than in the No-LS group, and higher scores of pain VAS and GJL were associated with a higher prevalence rate of LS. Therefore, we need to develop LS interventions for young adults, including the early screening of musculoskeletal symptoms and LS health education to prevent the mobility limitations of LS in the future.

## Figures and Tables

**Figure 1 healthcare-11-00532-f001:**
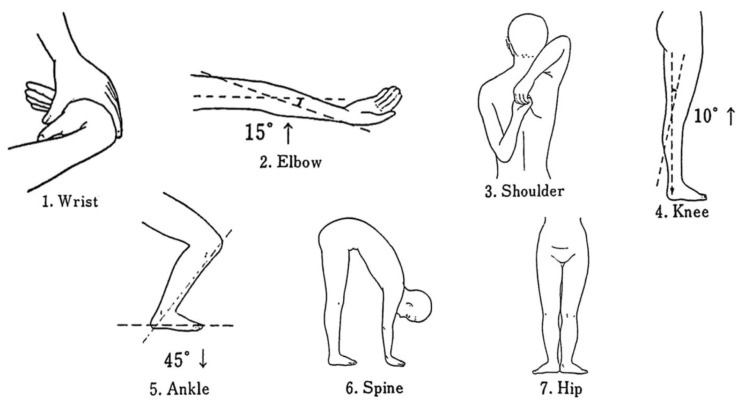
The University of Tokyo joint laxity test. Laxity of six major joints in the body (wrist, elbow, shoulder, knee, ankle, hip) and of the spine was examined.

**Figure 2 healthcare-11-00532-f002:**
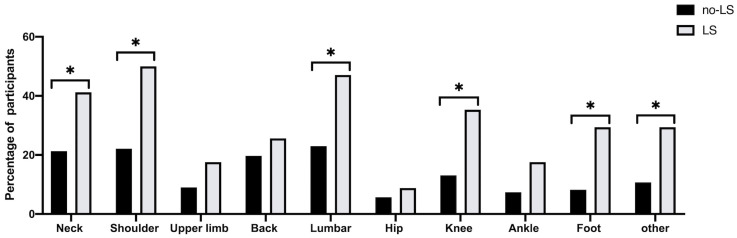
Prevalence of musculoskeletal pain in each anatomical area in No-LS and LS groups. * *p* < 0.05.

**Figure 3 healthcare-11-00532-f003:**
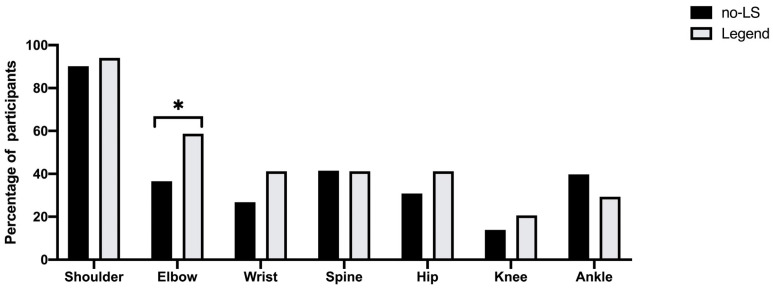
Prevalence of joint laxity risk sites in each anatomical area in No-LS and LS groups. * *p* < 0.05.

**Table 1 healthcare-11-00532-t001:** Anthropometric characteristics and background data of each group.

		Total (n = 157)	No-LS (n = 123)	LS (n = 34)	*p*-Value
Age (years)		19.8 ± 1.2	19.8 ± 1.0	20.0 ± 1.6	0.325
Sex (n, %)	Male	100 (63.7)	84 (68.3)	16 (47.1)	* 0.027
	Female	57 (36.3)	39 (31.7)	18 (52.9)	
Height (cm)		171.4 ± 8.1	171.7 ± 7.8	170.5 ± 9.3	0.449
Weight (kg)		63.6 ± 12.0	63.7 ± 11.8	63.5 ± 12.8	0.955
BMI (kg/m^2^)		21.6 ± 3.0	21.6 ± 3.0	21.7 ± 3.0	0.795
Total number of pain sites (n)		1.8 ± 2.3	1.4 ± 2.1	3.1 ± 2.6	*** <0.001
Pain VAS		3.92 ± 6.01	2.81 ± 4.96	7.94 ± 7.64	*** <0.001
GJL score		2.63 ± 1.49	2.46 ± 1.44	3.27 ± 1.50	** 0.005
Grip strength (kg)		30.6 ± 8.4	31.5 ± 8.1	27.7 ± 9.1	* 0.022
Grip strength/weight		0.48 ± 0.1	0.50 ± 0.1	0.43 ± 0.1	** 0.003
SMI (kg/m^2^)		7.90 ± 1.51	7.93 ± 1.62	7.79 ± 1.04	0.624
ASMI (kg/BMI × 100)		37.5 ± 4.8	38.0 ± 5.1	35.9 ± 3.1	* 0.030
PBF (%)		21.0 ± 6.6	20.2 ± 6.4	23.4 ± 6.8	* 0.012
TBW (%)		37.4 ± 7.6	37.9 ± 7.4	35.7 ± 8.0	0.129
Stand-up test (score)		7.4 ± 1.1	7.5 ± 0.9	7.0 ± 1.5	* 0.028
Two-step test (m/m)		1.54 ± 0.2	1.55 ± 0.2	1.53 ± 0.1	0.569
GLFS-25 (unit)		4.0 ± 4.9	2.2 ± 1.9	10.0 ± 7.0	*** <0.001
Usual gait speed (s/6 m)		0.78 ± 0.1	0.76 ± 0.1	0.81 ± 0.2	0.181
5xSST (s)		5.4 ± 1.0	5.4 ± 1.0	5.6 ± 0.9	0.261
IPAQ (Mets × min/wk)		1996.2 ± 1620.1	2009.2 ± 1513.8	1948.9 ± 1985.6	0.850
Sedentary time (min × weeks)		3662.4 ± 1104.6	3603.6 ± 1096.2	3867.2 ± 1129.8	0.223
Drinking (%)		48.0	46.7	52.9	0.734
Smoking (%)		1.3	1.6	0	1.000

1. Data are presented as means ± SD for age, height, weight, body mass index (BMI), total number of pain sites, pain visual analog scale (VAS), general joint laxity (GJL) score, grip strength, grip strength/weight, skeletal muscle mass/height2 (SMI), appendicular skeletal muscle mass/BMI*100 (ASMI), percent body fat (PBF), total body water (TBW), two-step test, stand-up test, 25-question geriatric locomotive function scale (GLFS-25), usual gait speed, five times sit-to-stand test (5xSST), international physical activity questionnaire (IPAQ) and sedentary time. Sex, drinking, and smoking habits are in percentages. 2. * *p* < 0.05; ** *p* < 0.01; *** *p* < 0.001. 3. No-LS: locomotive syndrome stage 0; LS: locomotive syndrome stage 1 or 2.

**Table 2 healthcare-11-00532-t002:** Logistic regression analysis of locomotive syndrome and pain sites.

Pain Sites	No-LS Group n (%)	LS Group n (%)	Unadjusted		Adjusted Model			
			(Crude)		(Model 1)		(Model 2)	
			OR (95% CI)	*p*-Value	OR (95% CI)	*p*-Value	OR (95% CI)	*p*-Value
Neck	26 (21.3%)	14 (41.2%)	2.59 (1.52–5.80)	* 0.021	2.13 (0.92–4.93)	0.076	2.53 (0.10–6.41)	0.051
Shoulder	27 (22.1%)	17 (50.0%)	3.52 (1.59–7.80)	** 0.002	3.11 (1.38–7.04)	** 0.006	2.99 (1.18–7.59)	* 0.021
Upper limb	11 (9.0%)	6 (17.6%)	2.16 (0.74–6.35)	0.161	2.00 (0.66–6.11)	0.223	2.53 (0.73–8.77)	0.144
Back	24 (19.7%)	9 (26.5%)	1.47 (0.61–3.56)	0.392	1.26 (0.51–3.13)	0.619	1.49 (0.54–4.10)	0.444
Lumbar	28 (23.0%)	16 (47.1%)	2.99 (1.35–6.61)	** 0.007	2.57 (1.14–5.82)	* 0.023	2.93 (1.19–7.21)	* 0.019
Hip	7 (5.7%)	3 (8.8%)	1.59 (0.39–6.51)	0.519	1.31 (0.30–5.62)	0.718	1.92 (0.37–9.98)	0.440
Knee	16 (13.1%)	12 (35.3%)	3.61 (1.50–8.70)	** 0.004	2.97 (1.20–7.38)	* 0.019	3.19 (1.16–8.73)	* 0.024
Ankle	9 (7.4%)	6 (17.6%)	2.69 (0.88–8.19)	0.081	2.60 (0.83–8.16)	0.102	4.14 (0.10–17.3)	0.051
Foot	10 (8.2%)	10 (29.4%)	4.67 (1.75–12.4)	** 0.002	4.41 (1.61–12.1)	** 0.004	6.18 (1.95–19.6)	** 0.002
Other	13 (10.7%)	10 (29.4%)	3.49 (1.37–8.90)	** 0.009	3.34 (1.23–8.67)	* 0.013	3.64 (1.27–10.6)	* 0.017

Notes: 1. Crude was the unadjusted model; Model 1 adjusted for [age (continuous variable) and sex (male, female); Model 2 included Model 1 plus body mass index (continuous variable), grip strength/weight (continuous variable), appendicular skeletal muscle mass/BMI*100 (continuous variable), percent body fat (continuous variable), general joint laxity score (continuous variable), international physical activity questionnaire (continuous variable), current smoking (no = 0, yes = 1), and current drinking (no = 0, yes = 1). 2. * *p* < 0.05; ** *p* < 0.01.

**Table 3 healthcare-11-00532-t003:** Logistic regression analysis of locomotive syndrome and pain VAS.

Pain Sites	No-LS Group	LS Group	Unadjusted		Adjusted Model			
			(Crude)		(Model 1)		(Model 2)	
			OR (95% CI)	*p*-Value	OR (95% CI)	*p*-Value	OR (95% CI)	*p*-Value
Total score	2.81 ± 4.96	7.94 ± 7.64	1.14(1.06–1.21)	*** 0.001	1.13(1.05–1.21)	** 0.001	1.13 (1.05–1.22)	** 0.002
Neck	0.40 ± 0.94	1.41 ± 2.23	1.56 (1.19–2.03)	** 0.001	1.48 (1.12–1.94)	** 0.005	1.57 (1.13–2.19)	** 0.008
Shoulder	0.42 ± 0.99	1.32 ± 1.63	1.69 (1.25–2.29)	*** 0.001	1.62 (1.19–2.20)	** 0.002	1.67 (1.15–2.42)	** 0.008
Upper limb	0.17 ± 0.66	0.53 ± 1.54	1.39 (0.96–2.03)	0.084	1.43 (0.99–2.07)	0.056	1.46 (0.97–2.18)	0.070
Back	0.37 ± 0.92	0.53 ± 1.08	1.17 (0.82–1.68)	0.390	1.08 (0.74–1.58)	0.687	1.09 (0.74–1.61)	0.671
Lumbar	0.57 ± 1.47	1.32 ± 1.77	1.30 (1.04–1.62)	* 0.022	1.24 (0.99–1.55)	0.060	1.24 (0.98–1.57)	0.073
Hip	0.12 ± 0.62	0.50 ± 0.09	1.07 (0.59–1.95)	0.835	0.94 (0.51–1.74)	0.838	1.00 (0.54–1.88)	0.990
Knee	0.25 ± 0.87	0.62 ± 1.13	1.41 (0.98–2.01)	0.062	1.32 (0.90–1.91)	0.153	1.62 (0.99–2.67)	0.054
Ankle	0.14 ± 0.55	0.56 ± 1.64	1.51 (1.00–2.28)	0.048	1.55 (1.00–2.41)	0.050	1.80 (1.04–3.11)	* 0.034
Foot	0.09 ± 0.32	0.62 ± 1.23	3.16 (1.60–6.68)	** 0.002	3.07 (1.43–6.59)	** 0.004	3.55 (1.47–8.56)	** 0.005
Other	0.31 ± 1.37	0.88 ± 2.19	1.19 (0.98–1.46)	0.091	1.17 (0.95–1.45)	0.149	1.17 (0.93–1.47)	0.187

Notes: 1. Crude was the unadjusted model; Model 1 adjusted for [age (continuous variable) and sex (male, female); Model 2 included Model 1 plus body mass index (continuous variable), grip strength/weight (continuous variable), appendicular skeletal muscle mass/BMI*100 (continuous variable), percent body fat (continuous variable), general joint laxity score (continuous variable), international physical activity questionnaire (continuous variable), current smoking (no = 0, yes = 1), and current drinking (no = 0, yes = 1). 2. ** p* < 0.05; ** *p* < 0.01, *** *p* < 0.001.

**Table 4 healthcare-11-00532-t004:** Logistic regression analysis of locomotive syndrome and joint laxity risk sites.

	No-LS Group n (%)	LS Group n (%)	Unadjusted		Adjusted Model			
			(Crude)		(Model 1)		(Model 2)	
			OR (95% CI)	*p*-Value	OR (95% CI)	*p*-Value	OR (95% CI)	*p*-Value
Wrist	33 (26.8%)	14 (41.2%)	1.91 (0.87–4.21)	0.109	1.70 (0.75–3.84)	0.204	1.79 (0.67–4.78)	0.245
Elbow	45 (36.6%)	20 (58.5%)	2.48 (1.14–5.38)	* 0.022	2.06 (0.91–4.60)	0.083	1.93 (0.74–5.01)	0.179
Shoulder	111 (90.2%)	32 (94.1%)	1.73 (0.37–8.13)	0.448	1.45 (0.30–7.02)	0.641	2.24 (0.20–25.1)	0.513
Knee	17 (13.9%)	7 (20.6%)	1.60 (0.60–4.25)	0.345	1.84 (0.67–5.06)	0.235	1.33 (0.39–4.55)	0.652
Ankle	49 (39.8%)	10 (29.4%)	0.63 (0.28–1.43)	0.269	0.63 (0.27–1.46)	0.280	0.74 (0.29–1.88)	0.525
Spine	51 (41.5%)	14 (41.2%)	0.99 (0.46–2.14)	0.976	0.85 (0.38–1.89)	0.683	0.88 (0.34–2.27)	0.789
Hip	38 (30.9%)	14 (41.2%)	1.57 (0.72–3.43)	0.262	1.51 (0.68–3.36)	0.314	1.59 (0.62–4.09)	0.339

Notes: 1. Crude was the unadjusted model; Model 1 adjusted for [age (continuous variable) and sex (male, female); Model 2 included Model 1 plus body mass index (continuous variable), grip strength/weight (continuous variable), appendicular skeletal muscle mass/BMI*100 (continuous variable), percent body fat (continuous variable), total pain VAS (continuous variable), international physical activity questionnaire (continuous variable), current smoking (no = 0, yes = 1), and current drinking (no = 0, yes = 1). 2. * *p* < 0.05.

**Table 5 healthcare-11-00532-t005:** Logistic regression analysis of locomotive syndrome and general joint laxity score.

	No-LS Group	LS Group n (%)	Unadjusted		Adjusted Model			
			(Crude)		(Model 1)		(Model 2)	
			OR (95% CI)	*p*-Value	OR (95% CI)	*p*-Value	OR (95% CI)	*p*-Value
General joint laxity score	2.46 ± 1.44	3.27 ± 1.50	1.44 (1.11–1.86)	** 0.006	1.38 (1.05–1.80)	* 0.019	1.42 (1.06–1.91)	* 0.019

Notes: 1. Crude was the unadjusted model; Model 1 adjusted for [age (continuous variable) and sex (male, female); Model 2 included Model 1 plus body mass index (continuous variable), grip strength/weight (continuous variable), appendicular skeletal muscle mass/BMI*100 (continuous variable), percent body fat (continuous variable), total pain VAS (continuous variable), international physical activity questionnaire (continuous variable), current smoking (no = 0, yes = 1), and current drinking (no = 0, yes = 1). 2. * *p* < 0.05; ** *p* < 0.01.

## Data Availability

Data is unavailable due to privacy or ethical restrictions.

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
