# Peer review of "Relationship between Locomotive Syndrome and Musculoskeletal Pain and Generalized Joint Laxity in Young Chinese Adults"

_healthcare, 2023, doi:10.3390/healthcare11040532_

Round 1

Reviewer 1 Report

REVIEW

I congratulate the Authors for the work done.

The manuscript is well-written, and the topic is current and interesting.

The scientific and practical value of the submitted work is high. 

However, I have a few comments/observations to consider:

1.  Material and Methods section:

- when assessing the condition of the musculoskeletal system and the occurrence of musculoskeletal pain, it is worth specifying the field of study of the respondents;

- did the exclusion criteria include a history of musculoskeletal injury? Injuries may have affected test results;

- it was not specified whether the respondents practiced sports professionally, which could have influenced the test results;

- why wasn't a more standardized test, the Beighton test, used to assess GJL?

Reviewer 2 Report

Comment to the authors

Introduction

The introduction is brief and clear. The objectives of the study are also clearly stated

Methodology

Participants

Line 61: Th acronym PE must be defined for readers to appreciate what the authors are referring to.

Inclusion and exclusion criteria are clearly defined.

2.4 Musculoskeletal pain

Line 115: “Musculoskeletal pain was defined by asking questionnaire about nine anatomical…”

The word questionnaire should be replaced with questions to read “Musculoskeletal pain was defined by asking questions about nine anatomical…”

Discussion

Line 228: It is not clear what comparison the authors are making by the sentence And the prevalence was higher in men < 29 years old than in those 30-50s.” giving that the article referenced has characterized the age group into “30-39”; “40-49” ; “49-59” etc. This should be clarified.

Reviewer 3 Report

Locomotive Syndrome Associated with Musculoskeletal Pain 2 and Generalized Joint Laxity in Young Chinese Adults

Introduction:

The introduction in general terms explains the incidence of LS, mostly studied in older people.

The study shows the prevalence of LS in young people. The introduction should provide more information of LS on this population group to justify the study.

I suggest to the authors that the title should be revised, using terms such as categorization, relationship between, detection etc.

Materials and Methods

Participants:

Participants with gait problems are excluded. One of the risk factors for SL is the presence of alterations in the gait pattern and foot position. Is your exclusion linked to not being able to conclude all the tests only? Or do the authors refer to orthopedic problems?

The data collection should be better explained e.g. in what year was the data collection done?

Was the survey self-administered? Was the entire data collection process done on the same day or in phases?

Is it possible to intuit that on the basis of these inclusion and exclusion criteria the participants were potentially healthy?

Physical Measurement:

The authors assess gait speed with a 6-meter test. On what procedure are they based to opt for this distance and not 10 meters as in other studies on this subject and same population.

It would be interesting to know the reference of the procedure.

The tests used are well known in the population over 60 years of age. Are they as reliable for this population group or have the same reference values been used as for the elderly?

GLFS-25:

The authors state that the JOA recommends this test also for young people. Can the authors provide the reliability of this test for young people?

This type of questionnaire is mostly for people over 60 years old and assesses very simple activities if the sample is potentially healthy and in a university environment.

On line 236 the authors discuss the results of the study with a pandemic justification.

It would be interesting to take this environment into account in the introduction of the study, as it can be considered as a risk factor.

Bibliography

The vast majority of the references are from studies with older people and we are dealing with an intervention with a young population.

Round 2

Reviewer 3 Report

The authors have satisfactorily complied with the reviewer's recommendations.

They are committed to continue research in this line to improve certain aspects of this article.